# THE 3D-PC: A BENCHMARK FOR VISUAL PERSPECTIVE TAKING IN HUMANS AND MACHINES

**Drew Linsley**[†1]**, Peisen Zhou**[†1]**, Alekh Karkada**[†1]**, Akash Nagaraj**[1]**,**
**Gaurav Gaonkar**[1]**, Francis E Lewis**[1]**, Zygmunt Pizlo**[2]**, Thomas Serre**[1]

`drew_linsley@brown.edu`

## ABSTRACT

Visual perspective taking (VPT) is the ability to perceive and reason about the perspectives of others. It is an essential feature of human intelligence, which develops over the first decade of life and requires an ability to process the 3D structure of visual scenes. A growing number of reports have indicated that deep neural networks (DNNs) become capable of analyzing 3D scenes after training on large image datasets. We investigated if this emergent ability for 3D analysis in DNNs is sufficient for VPT with the *3D perception challenge* (`3D-PC`): a novel benchmark for 3D perception in humans and DNNs. The `3D-PC` is comprised of three 3D-analysis tasks posed within natural scene images: **1.** a simple test of object *depth order*, **2.** a basic VPT task (*VPT-basic*), and **3.** another version of VPT (*VPT-Strategy*) designed to limit the effectiveness of "shortcut" visual strategies. We tested human participants (N=33) and linearly probed or text-prompted over 300 DNNs on the challenge and found that nearly all of the DNNs approached or exceeded human accuracy in analyzing object *depth order*. Surprisingly, DNN accuracy on this task correlated with their object recognition performance. In contrast, there was an extraordinary gap between DNNs and humans on *VPT-basic*. Humans were nearly perfect, whereas most DNNs were near chance. Fine-tuning DNNs on *VPT-basic* brought them close to human performance, but they, unlike humans, dropped back to chance when tested on *VPT-Strategy*. Our challenge demonstrates that the training routines and architectures of today's DNNs are well-suited for learning basic 3D properties of scenes and objects but are ill-suited for reasoning about these properties as humans do. We release our `3D-PC` datasets and code to help bridge this gap in 3D perception between humans and machines.

## 1 INTRODUCTION

In his theory of cognitive development, Piaget posited that human children gain the ability to predict which objects are visible from another viewpoint before the age of 10 (Piaget et al., 1956; Frick et al., 2014). This "Visual Perspective Taking" (VPT) ability is a foundational feature of human intelligence and a behavioral marker for the theory of mind (Aichhorn et al., 2006). VPT is also critical for safely navigating through the world and socializing with others (Fig. 1A). While VPT has been a focus of developmental psychology research since its initial description (Piaget et al., 1956; Bukowski, 2018; Martin et al., 2019) (Fig. 1B), it has not yet been studied in machines.

One of the more surprising results in deep learning has been the number of concomitant similarities to human perception exhibited by deep neural networks (DNNs), trained on large-scale static image datasets (Yamins & DiCarlo, 2016; Yamins et al., 2014). For example, DNNs now rival or surpass human recognition performance on object recognition and segmentation tasks (Geirhos et al., 2021; Linsley* et al., 2021; Lee et al., 2017), and are the state-of-the-art approach for predicting human neural and behavioral responses to images (Serre, 2019). There is also a growing number of reports indicating that DNNs trained with self-supervision or for object classification learn to encode 3D

---

[†]These authors contributed equally to this work.
[1]Carney Institute for Brain Science, Brown University, Providence, RI.
[2]Department of Cognitive Sciences, University of California-Irvine, Irvine, CA.

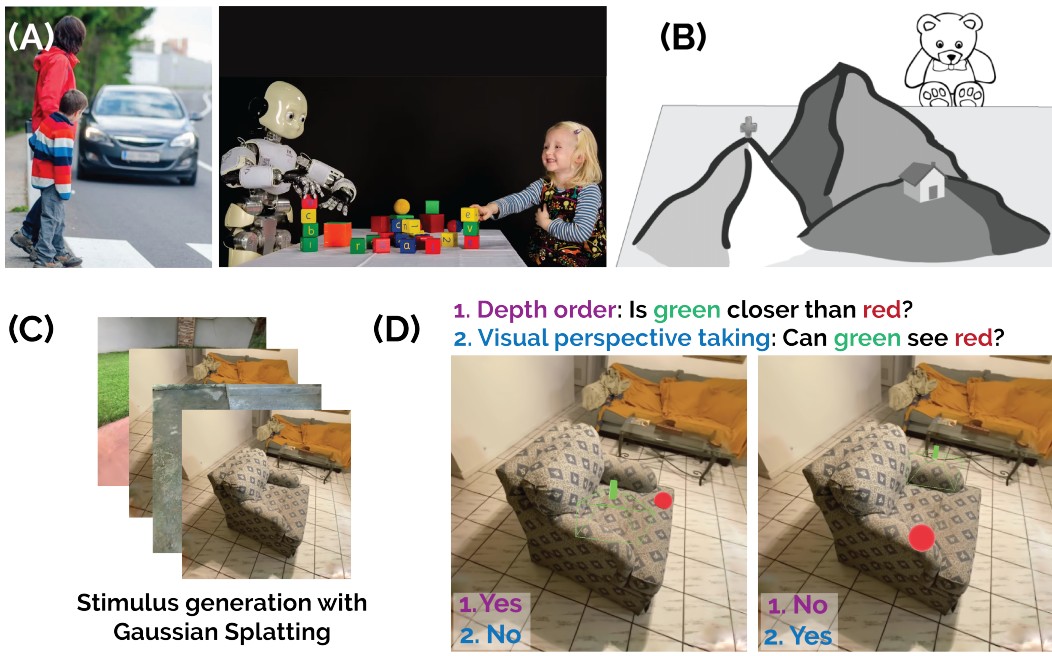

Figure 1: **Visual Perspective Taking (VPT) is the ability to analyze scenes from different viewpoints. (A)** Humans rely on VPT to anticipate the behavior of others. We expect that this ability will be essential for creating the next generation of AI assistants that can accurately anticipate human behavior (images are CC BY-NC). **(B)** VPT has been studied in developmental psychology since the mid-20th century using cartoon or highly synthetic stimuli. For example, Piaget's "Three Mountains Task" asks observers to describe the scene from the perspective of a bear (image from Bruce et al. (2017)). **(C)** Here, we use Gaussian Splatting (Kerbl et al., 2023) to develop a 3D scene generation pipeline for the *3D perception challenge* (3D-PC), to systematically compare 3D perception capabilities of human and machine vision systems. **(D)** The 3D-PC tests **1.** Object depth perception, and **2.** VPT.

properties of objects and scenes that humans are also sensitive to, such as the depth and structure of surfaces (Li et al., 2023; Ranftl et al., 2022; Saxena et al., 2023; Liu et al., 2023; Goodwin et al., 2022; Tang et al., 2023; Amir et al., 2021; Chen et al., 2023; Bhattad et al., 2023; El Banani et al., 2024). Are the emergent capabilities of DNNs for 3D vision sufficient for solving VPT tasks?

Here, we introduce the *3D perception challenge* (3D-PC) to address this question and systematically compare 3D perceptual capabilities of humans and DNNs. The 3D-PC evaluates observers on (Fig 1): **1.** identifying the order of two objects in depth (*depth order*), **2.** predicting if one of two objects can "see" the other (*VPT-basic*), and **3.** another version of VPT that limits the effectiveness of "shortcut" solutions (Geirhos et al., 2020a) (*VPT-Strategy*). The 3D-PC is distinct from existing psychological paradigms for evaluating VPT (Piaget et al., 1956; Bukowski, 2018; Martin et al., 2019) and computer vision challenges for 3D perception (El Banani et al., 2024; Amir et al., 2021) in two ways. First, unlike small-scale psychology studies of VPT, the 3D-PC uses a novel "3D Gaussian Splatting" (Kerbl et al., 2023) approach which permits the generation of endless real-world stimuli. Second, unlike existing computer vision challenges, our approach for data generation means that the 3D-PC tests and counterbalances labels for multiple 3D tasks on the exact same images, which controls for potential confounds in analysis and interpretation. We expect that DNNs which rival humans on the 3D-PC will become ideal models for a variety of real-world applications where machines must anticipate human behavior in real-time, as well as for enriching our understanding of how brains work (Fig. 1A).

**Contributions.** We built the 3D-PC and used it to evaluate 3D perception for human participants and 327 different DNNs. The DNNs we tested represented each of today's leading approaches, from

Visual Transformers (ViT) (Dosovitskiy et al., 2021a) trained on ImageNet-21k (Ridnik et al., 2021) to ChatGPT4 (Achiam et al., 2023) and Stable Diffusion 2.0 (Rombach et al., 2021).

- We found that DNNs were very accurate at determining the *depth order* of objects after linear probing or text-prompting. DNNs that are state-of-the-art on object classification matched or exceeded human accuracy on this task.

- However, DNNs dropped close to chance accuracy on *VPT-basic*, whereas humans were nearly flawless at this task.

- Fine-tuning the zoo of DNNs on *VPT-basic* boosted their performance to near human level. However, the performance of the DNNs — but not humans — dropped back to chance on *VPT-Strategy*. DNNs overly rely on the size and location features of objects instead of estimating their line-of-sight to solve VPT like humans do.

- Our findings demonstrate that the visual strategies necessary for solving VPT do not emerge in DNNs from large-scale static image training or after directly fine-tuning on the task. We release the `3D-PC` data, 3D Gaussian Splatting models, code, and human psychophysics to support the development of models that can perceive and reason about the 3D world like humans.

## 2    RELATED WORK

**3D perception in humans.**    The visual perception of 3D properties is a fundamentally ill-posed problem (Todd, 2004; Pizlo, 2010), which forces biological visual systems to rely on a variety of assumptions to decode the structure of objects and scenes. For example, variations in the lighting, texture gradients, retinal image disparity, and motion of an object all contribute to the perception of its 3D shape. 3D perception is further modulated by top-down beliefs about the structure of the world, which are either innate or shaped by prior sensory experiences, especially visual and haptic ones. In other words, humans learn about the 3D structure of the world in an embodied manner that is fundamentally different than how DNNs learn. In light of this difference, it would be remarkable if DNNs could accurately model how humans perceive their 3D world.

**Visual perspective taking in humans.**    VPT was devised to understand how capabilities for reasoning about objects in the world develop throughout the course of one's life. At least two versions of VPT have been introduced over the years (Michelon & Zacks, 2006; Pizlo, 2022). The version of VPT that we study here — known in the developmental literature as "VPT-1" — is the more basic form, which is thought to rely on automatic feedforward processing in the visual system (Michelon & Zacks, 2006). In light of the well-documented similarities between feedforward processing in humans and DNNs (Serre, 2019; Kreiman & Serre, 2020), we reasoned that this version of VPT would maximize the chances of success for today's DNNs.

**3D perception in DNNs.**    As deep neural networks (DNNs) have increased in scale and training dataset size over the past decade, their performance on essentially all visual challenges has improved. For example, there has also been significant progress in training DNNs to solve 3D computer vision tasks. For example, it is possible to train models that can precisely decode object and scene depth Yang et al. (2024), generate a high-resolution 3D model of an environment Kerbl et al. (2023), or even predict new camera views of an object given a single video of it Zhang et al. (2024). DNNs represent the state-of-the-art approach to nearly every 3D vision task today.

Surprisingly, there is also growing evidence that 3D perception emerges in DNNs even when they are trained on static image tasks. For example, DNNs trained with a variety of self-supervised learning techniques on static image datasets learn to represent the depth, surface normals, and 3D correspondence of features in scenes (Ranftl et al., 2022; Saxena et al., 2023; Liu et al., 2023; Goodwin et al., 2022; Tang et al., 2023; Amir et al., 2021; Chen et al., 2023; Bhattad et al., 2023; El Banani et al., 2024). While similarities between DNNs and human 3D perception have yet to be evaluated systematically, it has been shown that there are differences in how the two reason about the 3D shape of objects (Qian & Ullman, 2024). The `3D-PC` complements prior work by systematically evaluating which aspects of human 3D perception today's DNNs can and cannot accurately represent.

**Limitations of DNNs as models of human visual perception.**    Over recent years, DNNs have grown progressively more accurate as models of human vision for object recognition tasks (Geirhos

et al., 2021; 2020a). At the same time, these models which succeed as models of human object recognition struggle to capture other aspects of visual perception (Bowers et al., 2022) including contextual illusions (Linsley et al., 2020), perceptual grouping (Kim* et al., 2020; Linsley et al., 2021), and categorical prototypes (Golan et al., 2020). There are also multiple reports showing that DNNs are growing less aligned with the visual strategies of humans and non-human primates as they improve on computer vision benchmarks (Linsley et al., 2023b; Fel* et al., 2022; Linsley et al., 2023a). The `3D-PC` provides another axis upon which the field can evaluate DNNs as models of human vision.

## 3 METHODS

**The `3D-PC`.** To enable a fair comparison between human observers' and DNNs' 3D perceptual capabilities, we designed the `3D-PC` framework with two goals: **1.** posing different 3D tasks on the same set of stimuli, and **2.** generating a large number of stimuli to properly train DNNs on these tasks. We achieved these goals by combining 3D Gaussian Splatting (Kerbl et al., 2023), videos from the Common Objects in 3D (Co3D) (Reizenstein et al., 2021) dataset, and Unity (Juliani et al., 2018; Pranckevicius, 2023) into a flexible data-generating framework.

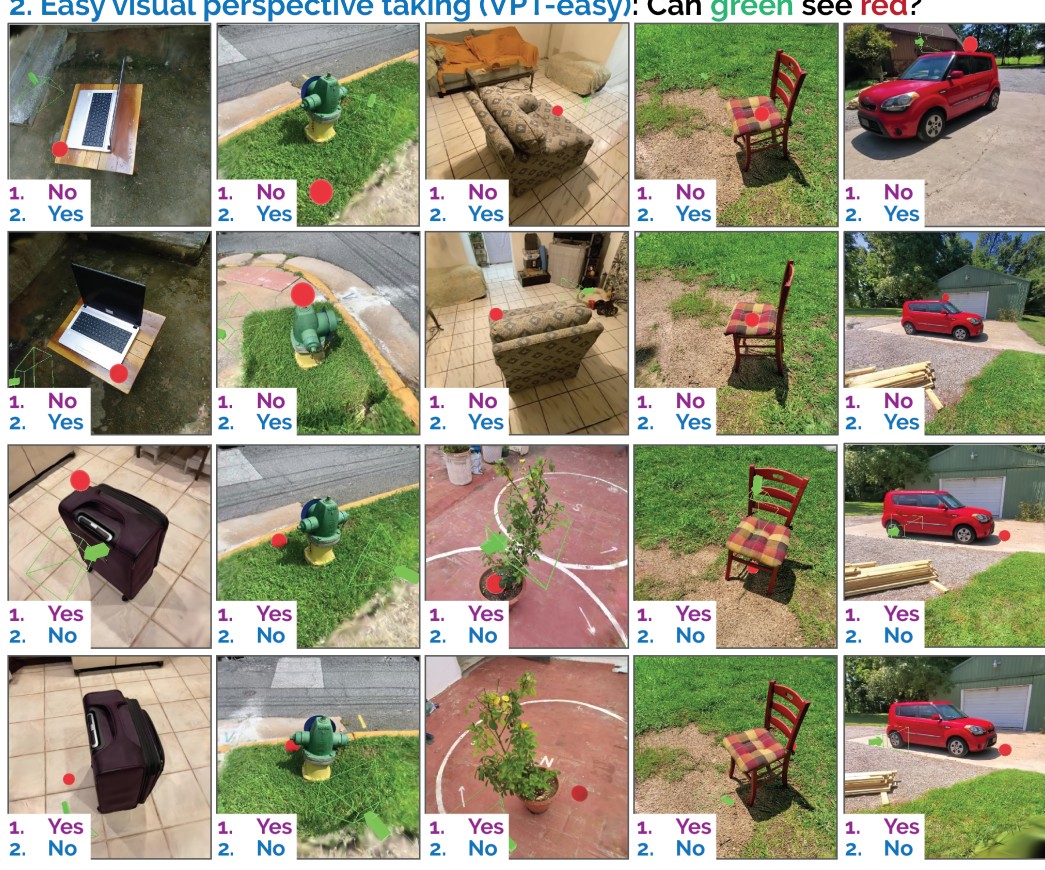

Figure 2: **`3D-PC` examples.** We tested 3D perception in images generated by Gaussian Splatting. Each image depicts a green camera and a red ball. These objects are placed in the scene in a way that counterbalances labels for depth order task and VPT-basic tasks.

Our procedure for building the `3D-PC` involved the following three steps. **1.** We trained Gaussian Splatting models on videos in Co3D (Fig. 1C). **2.** We imported these trained models into Unity, where we added green camera and red ball objects into each 3D scene, which were used to pose visual tasks (Fig. 1D). **3.** We then generated random viewpoint trajectories within each 3D scene,

rendered images at each position along the trajectory, and derived ground-truth answers for *depth order* and VPT tasks for the green camera at every position from Unity.

Our approach makes it possible to generate an unlimited number of visual stimuli that test an observer's ability to solve complementary 3D perception tasks (*depth order* and VPT) while keeping visual statistics constant and ground truth labels counterbalanced across tasks. For the version of 3D-PC used in our evaluation and released at `https://huggingface.co/datasets/3D-PC/3D-PC`, the *depth order* and *VPT-basic* tasks are posed on the same set of 7,480 training images of 20 objects and scenes, and a set of 94 test images of 10 separate objects and scenes (Fig. 2). We held out a randomly selected 10% of the training images for validation and model checkpoint selection.

To build the *VPT-Strategy* task, we rendered images where we fixed the scene camera while we moved the green camera and red ball objects to precisely change the line-of-sight between them from unobstructed to obstructed and back. We reasoned that this experiment would reveal if an observer adopts the visual strategy of taking the perspective of the green camera, which is thought to be used by humans (Michelon & Zacks, 2006), from other strategies that relied on less robust feature-based shortcuts. This dataset consisted of a test set of 100 images for 10 objects and scenes that were not included in *depth order* or *VPT-basic*.

**Psychophysics experiment.** We tested 10 participants on *depth order*, 20 on *VPT-basic*, and 3 on *VPT-Strategy*. 33 participants were recruited online from Prolific. All provided informed consent before completing the experiment and received $15.00/hr compensation for their time (this amounted to $5.00 for the 15–20 minutes the experiment lasted). These data were de-identified.

Participants were shown instructions for one of the 3D-PC tasks, then provided 20 training examples to ensure that they properly understood it (Appendix Fig 8). These human training examples were drawn from the DNN training set. Each experimental trial consisted of the following sequence of events overlaid onto a white background: **1.** a fixation cross displayed for 1000*ms*; **2.** an image displayed for 3000*ms*, during which time the participants were asked to render a decision. Participants pressed one of the left or right arrow keys on their keyboards to provide decisions.

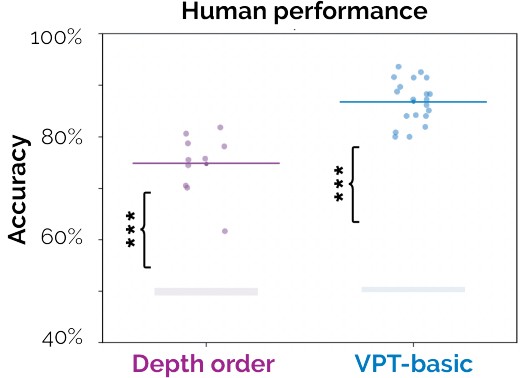

Figure 3: **Human accuracy for object depth order and VPT-basic tasks.** Bars near 50% are label-permuted noise floors; lines are group means. The difference is significant, *** = $p < 0.001$.

Images were displayed at 256×256 pixel resolution, which is equivalent to a stimulus between $5° - 11°$ of visual angle across the range of display and seating setups we expected our online participants used for the experiment.

**Model zoo.** We evaluated a wide range of DNNs on the 3D-PC, which represented the leading approaches for object classification, self-supervised pretraining, image generation, depth prediction, and vision language modeling (VLM). Our zoo includes 317 DNNs from PyTorch Image Models (TIMM) (Wightman, 2019), ranging from classic models like AlexNet (Krizhevsky et al., 2012a) to state-of-the-art models like EVA-02 (Fang et al., 2023) (see Appendix 1 for the complete list). We added foundational vision models like MAE (He et al., 2022), DINO v2 (Oquab et al., 2023), iBOT (Zhou et al., 2021), SAM (Kirillov et al., 2023), and Midas (Ranftl et al., 2022) (obtained from the GitHub repo of El Banani et al. (2024)). We also included Depth Anything (Yang et al., 2024), a foundational model 3D scene analysis and depth prediction (El Banani et al., 2024), as well as the Stable Diffusion 2.0 (Rombach et al., 2021) image generation model. Finally, we added state-of-the-art large vision language models (VLMs) ChatGPT4 (Achiam et al., 2023), Gemini (Team et al., 2023), and Claude 3 (Anthropic, 2024). We evaluated a total of 327 models on the 3D-PC.

**Model evaluation.** We evaluated all models except for the VLMs on the *depth order* and *VPT-basic* tasks in this challenge by training linear probes on image embeddings from their penultimate layers.

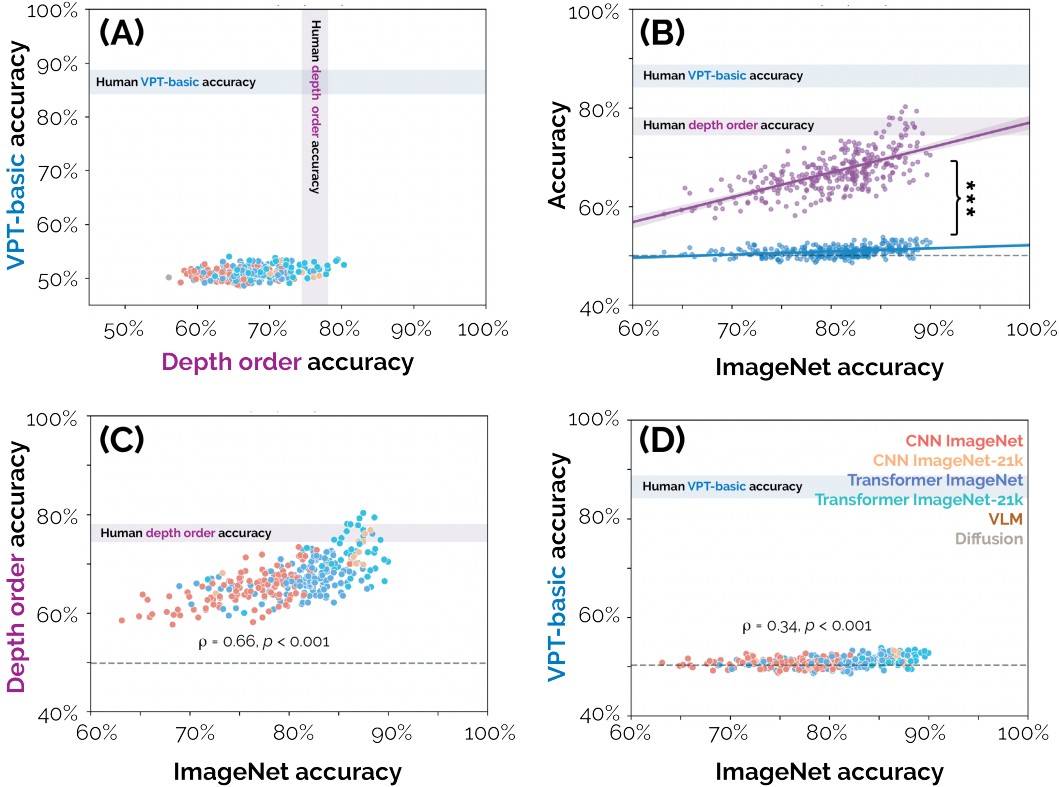

Figure 4: **DNN performance on the depth order and VPT-basic tasks in the `3D-PC` after *linear probing or prompting*.** **(A, B)** DNNs are significantly more accurate at depth order than *VPT-basic*. Human confidence intervals are S.E.M. and ***: $p < 0.001$. **(C, D)** DNN accuracy for *depth order* and *VPT-basic* strongly correlates with object classification accuracy on ImageNet. Dashed lines are the mean of label-permuted human noise floors.

Linear probes were trained using PyTorch (Paszke et al., 2019) for 50 epochs, a $5e$-4 learning rate, and early stopping (see Appendix A.5 for details). Training took approximately 20 minutes per model using NVIDIA-RTX 3090s. We tested the `Stable Diffusion 2.0` model by adopting the evaluation method used in Li et al. (2023) (see Appendix A.8 for details). We evaluated the VLMs by providing them the same instructions and training images (along with ground truth labels) given to humans, then recording their responses to images from each task via model APIs.

To test the learnability of the `3D-PC`, we also fine-tuned each of the TIMM models in our zoo to solve the tasks. To do this, we trained each of these models for 30 epochs, a $5e$-5 learning rate, and early stopping (see Appendix A.5 for details). Fine-tuning took between 3 hours and 24 hours per model using NVIDIA-RTX 3090s.

## 4    RESULTS

**Humans find VPT easier than determining the depth ordering of objects.**    All humans participants and models were test on the same 94 images, each of which had a VPT label and a depth label. Human participants were on average 74.73% accurate at determining the *depth order* of objects, and 86.82% accurate at solving the *VPT-basic* task (Fig. 3; $p < 0.001$ for both; statistical testing done through randomization tests (Edgington, 1964)). Humans were also significantly more accurate at solving *VPT-basic* than they were at the *depth order* task.

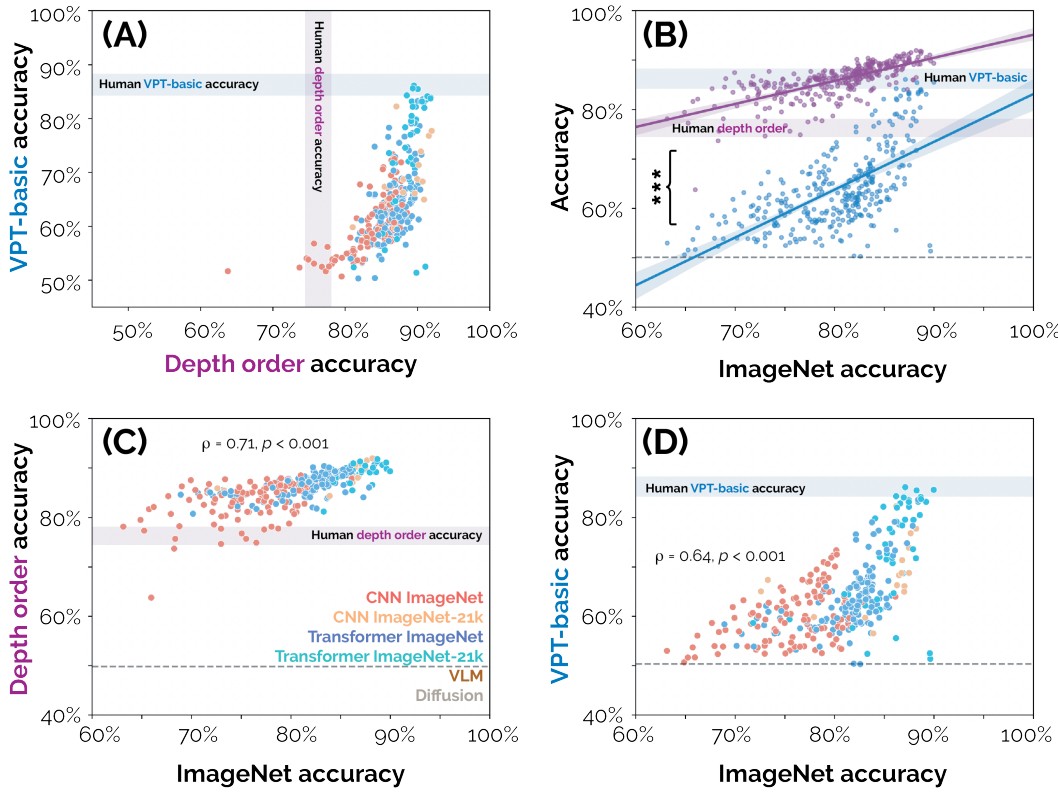

Figure 5: **DNN performance on the depth order and VPT-basic tasks in the `3D-PC` after *fine-tuning*.** (A) Fine-tuning makes DNNs far better than humans at the *depth order* task and improves the performance of several DNNs to be at or beyond human accuracy on *VPT-basic*. (B) Even after fine-tuning, there is still a significant difference in model performance on *depth order* and *VPT-basic* tasks, $p < 0.001$. (C, D) DNN accuracy on both tasks after fine-tuning correlates with ImageNet object classification accuracy. Human confidence intervals are S.E.M. and ***: p < 0.001. Dashed lines are the mean of label-permuted human noise floors.

**DNNs learn depth but not VPT from static image training.** DNNs showed the opposite pattern of results on *depth order* and *VPT-basic* tasks as humans after linear probing or prompting (Fig. 4): 15 of the DNNs we tested fell within the human accuracy confidence interval on the *depth order* task, and three even outperformed humans (Fig. 4A). In contrast, while humans were on average 86.82% accurate at *VPT-basic*, the DNN which performed the best on this task, the ImageNet 21K-trained `beit` (Bao et al., 2021a), was 53.82% accurate. Even commercial VLMs struggled on *VPT-basic* and were around chance accuracy (`ChatGPT4`: 52%, `Gemini`: 52%, and `Claude 3`: 50%). The *depth order* task was significantly easier for DNNs than *VPT-basic* ($p < 0.001$), which is the opposite of humans (Fig. 4B).

**ImageNet accuracy correlates with the 3D capabilities of DNNs.** What drives the development of 3D perception in DNNs trained on static images? We hypothesized that as DNNs scale up, they learn ancillary strategies for processing natural images, including the ability to analyze the 3D structure of scenes. To investigate this possibility, we focused on the TIMM models in our DNN zoo. These models have previously been evaluated for object classification accuracy on ImageNet, which we used as a stand-in for DNN scale (Fel* et al., 2022; Linsley et al., 2023a;b). Consistent with our hypothesis, we found a strong and significant correlation between DNN performance on ImageNet and *depth order* task accuracy ($\rho = 0.66, p < 0.001$, Fig. 4C). Despite the very low accuracy of DNNs on *VPT-basic*, there was also a weaker but still significant correlation between performance

on this task and ImageNet ($\rho = 0.34$, $p < 0.001$, the difference in correlations between the tasks is $\rho = 0.32$, $p < 0.001$; Fig. 4D). These results suggest that monocular depth cues develop in DNNs alongside their capabilities for object classification[1]. However, the depth cues that DNNs learn are poorly suited for VPT.

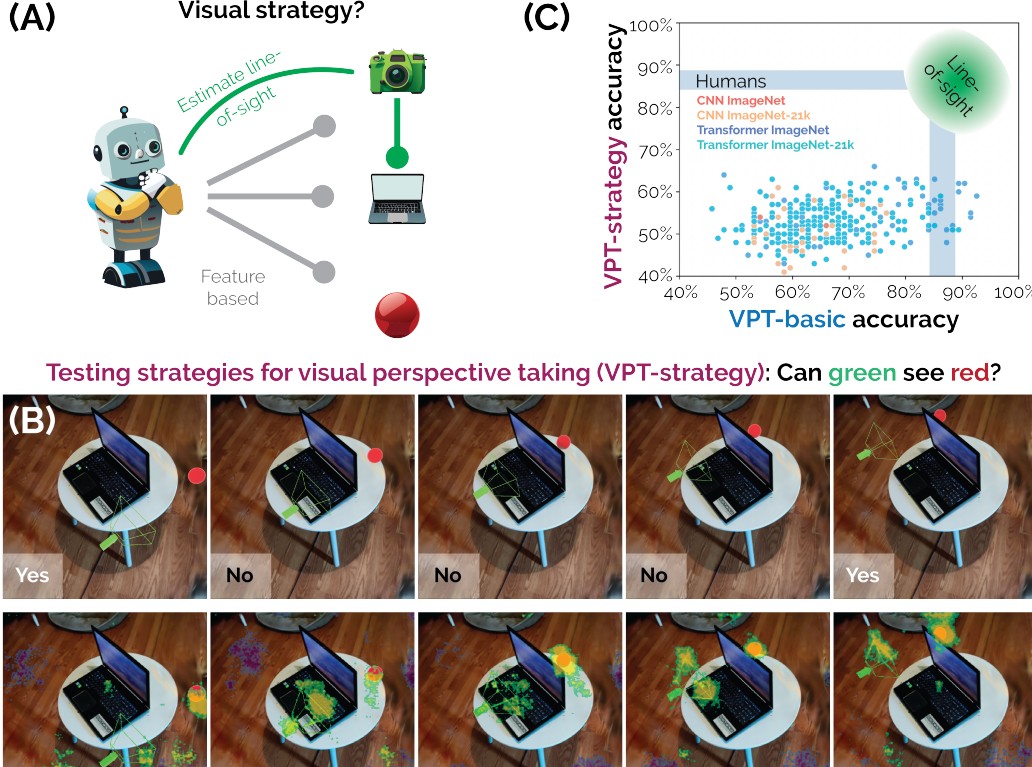

Figure 6: **Even DNNs fine-tuned on VPT-basic fail on VPT-Strategy.** **(A)** To better characterize the strategy used by humans and DNNs to solve VPT, we devised a new test, *VPT-Strategy*, in which the green camera and red ball are moved through a scene while holding the scene camera and a centrally-positioned object still. This task is easily solvable if an observer *estimates the line-of-sight* of the green camera; other strategies, such as those that rely on specific image features like the relative sizes and positions of objects (*feature based*), may be less effective. **(B)** Examples of *VPT-Strategy* stimuli along with the ground-truth label (top-row) and predictions by a `ViT large` after linearly probing or fine-tuning for *VPT-basic* (bottom-row). Decision attribution maps from each version of the `ViT large`, derived from "smooth gradients" (Smilkov et al., 2017), are overlaid onto bottom-row images (purple/blue=linearly probed, yellow/green=fine-tuned). The fine-tuned ViT locates the green camera and red ball but renders incorrect decisions. **(C)** DNNs fine-tuned on *VPT-basic* fail to solve *VPT-Strategy*; they rely on a brittle feature-based strategy. Humans, on the other hand, are 87% accurate; they likely estimate line-of-sight.

**DNNs can solve *VPT-basic* after fine-tuning.**    One possible explanation for the failure of today's DNNs on *VPT-basic* is that the task requires additional cues for 3D vision that cannot be easily learned from static images. To explore this possibility, we fine-tuned each of the TIMM models in our DNN zoo to solve *depth order* and *VPT-basic* (Fig. 5A). There was still a significant difference between DNN performance on the two tasks (Fig. 5B, $p < 0.001$), but fine-tuning caused 97% of the

---

[1]More work is needed to identify a causal relationship between the development of monocular depth cues and object recognition accuracy.

DNNs to exceed human accuracy on *depth order*, and four of the DNNs to reach human accuracy on *VPT-basic*. DNN performance on the tasks more strongly correlated with ImageNet accuracy after fine-tuning than linear probing (compare Fig. 5C/D and Fig. 4C/D). We also compared the errors these DNNs made on both tasks to humans. We found nearly all of the fine-tuned DNNs were aligned with humans on *depth order*, and a handful were aligned with humans on *VPT-basic* (Fig. 9).

**DNNs learn different strategies than humans to solve VPT.**   Prior work has found that Deep Neural Networks (DNNs) can achieve human-level performance on visual tasks while using fundamentally different strategies than humans (Linsley et al., 2023b; Fel* et al., 2022; Linsley et al., 2023a). To investigate whether this issue applies to VPT, we designed a new experiment which we call *VPT-Strategy*.

*VPT-Strategy* was inspired by past work in developmental psychology, where it has been proposed that humans solve VPT by estimating line-of-sight (Fig. 6A) because humans respond in predictable ways when objects in a scene are slightly repositioned (Pizlo, 2022; Michelon & Zacks, 2006). In *VPT-Strategy*, observers solve the VPT task on a series of images rendered from a fixed camera viewpoint which show the green camera and red ball incrementally repositioned from one side of the screen to the other, passing by an occluding object in the process. This makes it possible to precisely map out the moments at which the green camera's view of the red ball is unoccluded, then occluded, then unoccluded once more. DNNs that have been fine-tuned on *VPT-basic* behaved differently than humans on this task: humans were 87% accurate, but the highest performing DNN, the `Swin Transformer` (Liu et al., 2021) trained on ImageNet-21k, was only 66% accurate (Fig. 6C). Since DNNs cannot generalize to *VPT-Strategy*, it means that they do not learn to estimate line-of-sight to solve VPT.

To better understand the source of DNN failures, we derived "smooth gradients" (Smilkov et al., 2017) decision attribution maps of every DNN that we tested on *VPT-Strategy*, and then measured the overlap of these maps with the green camera and red ball objects using Dice similarity coefficient (Dice, 1945). These attribution maps revealed that the DNNs we evaluated did indeed learn to attend to the locations of the green camera and red ball objects when trying to solve VPT (Fig. 6B and Fig. 10), and the precision of their localization increased as a function of accuracy on *VPT-basic* (Fig. 10). When considered alongside the poor DNN performance on *VPT-strategy*, these results imply that DNNs learn a brittle feature-based strategy that is dependent on object properties like size and location instead of estimating the line-of-sight of the green camera like humans do (Fig. 6C).

## 5  DISCUSSION

Deep neural networks (DNNs) have rapidly advanced over recent years to the point where they match or surpass human-level performance on numerous visual tasks. However, our `3D-PC` reveals there is still a significant gap between the abilities of humans and DNNs to reason about 3D scenes. While DNNs match or exceed human accuracy on the basic object *depth order* task after linear probing or prompting, they struggle remarkably on even the basic form of VPT that we test in the `3D-PC`. Fine-tuning DNNs on *VPT-basic* allows them to approach human-level performance, but unlike humans, their strategies do not generalize to the *VPT-Strategy* task.

A striking finding from our study is the strong correlation between DNNs' object classification accuracy on ImageNet and their performance on *depth order* and *VPT-basic*. This correlation suggests that monocular depth cues emerge in DNNs as a byproduct of learning to recognize objects, potentially because these cues are useful for segmenting objects from their backgrounds. The difference in DNN effectiveness for *depth order* versus *VPT-basic*, however, indicates that these cues are not sufficient for reasoning about the 3D structure of scenes in the way that VPT demands.

Thus, today's approaches for developing DNNs, which primarily focus on static image datasets, may be poorly suited for enabling robust 3D perception and reasoning abilities akin to those of humans. Incorporating insights from human cognition and neuroscience into DNNs, particularly in ways biological visual systems develop 3D perception, could help evolve more faithful models of human intelligence.

**Limitations**   One key limitation of our study is that we were not able to isolate the precise cues that humans versus DNNs use to solve *depth order*. While both were very capable of solving the

task, it's very possible that both relied on different monocular depth cues. For example, the relative size of objects is a very salient cue for solving the task. However, object occlusion and interposition is another strong cue in this dataset, and it's unclear if DNNs and humans rely on one or the other, or both, or other cues like the lighting and shadows of objects. Future work that can drill down on the strategies of humans and DNNs for perceiving 3D spatial properties will make significant contributions for understanding the alignment of biological and machine vision.

Another key limitation of our study is that our version of VPT represents the most basic form studied in the developmental psychology literature. While solving this task is evidently an extraordinary challenge for DNNs, it is only one small step towards human-level capabilities for reasoning about 3D worlds in general. Far more research is needed to identify additional challenges, architectures, and training routines that can help DNNs perceive and reason about the world like humans do. We release our `3D-PC` data, 3D Gaussian Splatting models, and code anonymously at `https://huggingface.co/datasets/3D-PC/3D-PC` to support this goal.

Finally, we validated developmental psychology work that humans rely on a line-of-sight strategy to solve VPT, and demonstrated in our *VPT-Strategy* task that DNNs learn to use a different strategy after fine-tuning. We provided evidence through our attribution map experiment (Fig. 10) that DNN strategies are highly focused on the target objects. However, we were not able to characterize exactly how DNNs use object features for VPT, which may be critical for developing the next-generation of vision models that can solve VPT like humans do.

## ACKNOWLEDGMENTS

Our work is supported by ONR (N00014-24-1-2026, NSF (IIS-2402875), the ANR-3IA Artificial and Natural Intelligence Toulouse Institute (ANR-19-PI3A-0004). Additional support was provided by the Carney Institute for Brain Science and the Center for Computation and Visualization (CCV). We acknowledge the Cloud TPU hardware resources that Google made available via the TensorFlow Research Cloud (TFRC) program as well as computing hardware supported by NIH Office of the Director grant S10OD025181.

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

# A APPENDIX

## A.1 AUTHOR STATEMENT

As authors of this dataset, we bear all responsibility for the information collected and in case of violation of rights and other ethical standards. We affirm that our dataset is shared under a Creative Commons CC-BY license.

## A.2 DATA ACCESS

We release benchmarking code, data, and 3D Gaussian Splatting models anonymously at `https://huggingface.co/datasets/3D-PC/3D-PC`.

## A.3 POTENTIAL NEGATIVE SOCIETAL IMPACTS OF THIS WORK

The most obvious potential negative impact of our work is that advancing visual perspective taking (VPT) capabilities in artificial agents could potentially enable militaristic applications or surveillance overreach. However, we hope that our benchmark will aid in the development of AI-based assistants that can better anticipate and react to human needs and social cues for safer navigation and interaction. We also believe that our benchmark will guide the development of better computational models of human 3D perception as well as the neural underpinnings of these abilities.

## A.4 DATA GENERATION

To generate data for the `3D-PC`, we first trained 3D Gaussian Splatting (Kerbl et al., 2023) models on videos from the Common Objects in 3D (Co3D) (Reizenstein et al., 2021), which yielded 3D representations of each scene. We then imported trained models into Unity (Juliani et al., 2018) using Unity Gaussian Splatting (Pranckevicius, 2023) and added 3D models of the green camera and red ball to each. Finally, we rendered 50 images along a smooth viewpoint camera trajectory sampled near the original trajectory used for training the Gaussian Splatting model. For each 3D scene, we created 5 positive and 5 negative settings for VPT.

To generate *VPT-basic*, the generation process was repeated for 30 Co3D videos from 10 different categories. We removed any images where the green camera and red ball were not visible. We then split the images into a training set of 7480 images from 20 scenes and a testing set of 94 images from 10 other scenes. For the *depth order* task, we used the same data splits but removed any ambiguous samples where the objects were similarly close to the camera. The resulting dataset for the *depth*

*order* task contains 4787 training images and 94 testing images. The same set of testing images is used for both model and human benchmarks.

For *VPT-Strategy*, we used the same process to generate data from 10 additional Co3D scenes not included in *VPT-basic* and additionally controlled the positions of the green camera and the red ball. The angle between these two objects was held constant while we moved them so that their line of sight was unobstructed, obstructed, and then unobstructed once again. For each Co3D scene, we rendered 10 settings from a fixed viewpoint camera position, resulting in 100 images in total for *VPT-Strategy*.

### A.5   MODEL ZOO

We linearly probed 317 DNNs from Pytorch Image Models (TIMM) (Wightman, 2019) (Table 1) along with foundational vision models following the procedures in (El Banani et al., 2024). All DNNs were trained and evaluated with NVIDIA-RTX 3090 GPUs. All linear probes were trained for 50 epochs, with a $5e-4$ learning rate, a $1e-4$ weight decay, a 0.3 dropout rate, and a batch size of 128. We fine-tuned each of the TIMM models for 30 epochs, a $5e-5$ learning rate, $1e-4$ weight decay, 0.7 dropout rate, and a batch size of 16. Linear probing took approximately 20 minutes per model, and fine-tuning varied from 3 to 24 hours on a NVIDIA-RTX 3090 GPU.

### A.6   VLM EVALUATION

We evaluated the following proprietary VLMs on the *VPT-basic* and *depth order* tasks: GPT-4 (`gpt-4-turbo`), Claude (`claude-3-opus-20240229`), and Gemini (`gemini-pro-vision`). To evaluate these VLMs, we used their APIs to send queries containing 20 training images, with ground truth answers as context, as well as a test image. The prepended 20 training images meant that for every example in the challenge, VLMs were given the opportunity to learn, "in-context", how to solve the given task.

The prompt we used for the depth task was "`In this image, is the red ball closer to the observer or is the green arrow closer to the observer? Answer only BALL if the red ball is closer, or ARROW if the green arrow is closer, nothing else.`" and the prompt for the *VPT-basic* task was "`In this image, if viewed from the perspective of the green 3D arrow in the direction the arrow is pointing, can a human see the red ball? Answer only YES or NO, nothing else`". We evaluated each model's generated responses across multiple temperatures, ranging from 0.0 to 0.7 in increments of 0.1, and we report the average of the best 3 runs. Note that while this evaluation approach gives the VLMs more opportunities to perform well on our benchmark than other models, they still struggled immensely (see main text).

### A.7   VLM CHAIN-OF-THOUGHT EVALUATION

Chain-of-Thought (COT) prompting is a technique used to improve VLM and LLM performance by showing models the conceptual steps that might be helpful for solving a problem. COT has proven very effective on reasoning tasks. To understand if COT would also help DNNs solve VPT, we devised a prompting approach described below (see Fig. 7 for an example image used when prompting for COT reasoning). However, COT did not improve VLM accuracy on VPT (it achieved chance accuracy).

> **Depth order:** Here is a sample training image, from the perspective shown, is the red ball closer to the observer or is the green arrow closer to the observer? Answer only 'BALL' if the red ball is closer, or 'ARROW' if the green arrow is closer, nothing else. To answer this, let's think step by step: The red ball is above the suitcase, which is a small distance away from the observer, approximately 3 feet. The green arrow is in front of the suitcase, which is closer to the observer, approximately 1 foot away. Thus the answer is: ARROW.

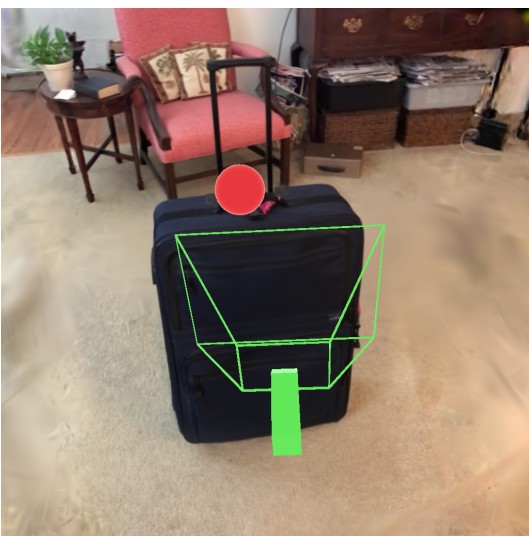

Figure 7: **An example image for Chain-of-Thought prompting of vision language models to solve VPT.**

> **VPT:** Here is a sample training image, if viewed from the perspective of the green 3D arrow, in the direction the arrow is pointing, can a human see the red ball? Answer only 'YES' or 'NO', nothing else. Let's think step by step: If a human were looking in the direction the arrow was pointing, they would face a few possible occlusions, such as the suitcase. The red ball is above the suitcase, so the suitcase wouldn't block the view of the red ball. Thus, the answer is: YES

## A.8 STABLE DIFFUSION EVALUATION

We followed the method of Li et al. (2023) to evaluate `Stable Diffusion 2.0` on the `3D-PC`. This involved trying multiple prompts to optimize the zero-shot classification performance of the `Stable Diffusion 2.0` model, on *VPT-basic* and *depth order* tasks. For *VPT-basic* we found that the prompt `"A photo with red ball is visible from the green arrow's perspective"` for positive class and `"A photo with red ball not visible from the green arrow's perspective"` for the negative class led to the best performance. For the *depth order* task, the prompt with the highest performance was `"A photo with green arrow closer to the camera as compared to red ball"` and `"A photo with red ball closer to the camera as compared to green arrow"` for positive and negative classes respectively.

## A.9 HUMAN BENCHMARK

We recruited 30 participants through Prolific, compensating each with $5 upon successful completion of all test trials. Participants confirmed their completion by pasting a unique system-generated code into their Prolific accounts. The compensation was prorated based on the minimum wage. We also incurred a 30% overhead fee per participant paid to Prolific. In total, we spent $195 on these benchmark experiments.

### A.9.1 EXPERIMENT DESIGN

At the outset of the experiment, we acquired participant consent through a form approved by the Institutional Review Board (IRB). The experiment was performed on a computer using the Chrome browser. Following consent, we presented a demonstration with instructions and an example video. Participants had the option to revisit the instructions at any time during the experiment by clicking a link in the top right corner of the navigation bar.

In the *depth order* task, the participants were asked to classify the image as "positive" (the green arrow in closer to the viewer) or "negative" (the red ball is closer) using the right and left arrow keys respectively. The choice for keys and their corresponding instances were mentioned below the image on every screen (See Appendix Fig. A1. Participants were given feedback on their response

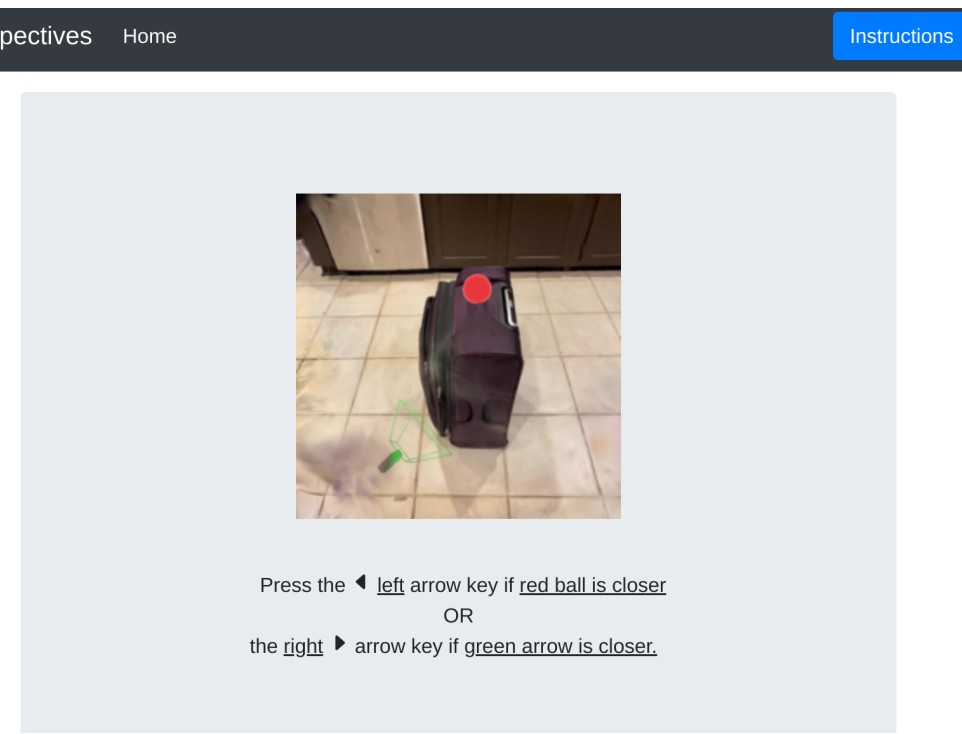

Figure 8: **An experiment trial.**

(correct/incorrect) during every practice trial, but not during the test trials. In the VPT tasks, the choices were "the green arrow/camera see the red ball" or "the green arrow/camera can not see the red ball".

The experiment was not time-bound, allowing participants to complete it at their own pace. Participants typically took around 20 minutes. After each trial, participants were redirected to a screen confirming the successful submission of their responses. They could start the next trial by clicking the "Continue" button or pressing the spacebar. If they did not take any action, they were automatically redirected to the next trial after 1000 milliseconds. Additionally, participants were shown a "rest screen" with a progress bar after every 40 trials, where they could take additional and longer breaks if needed. The timer was turned off during the rest screen.

### A.10    HUMAN VS. DNN DECISION MAKING ON *VPT-basic*

We compared the decision strategies of humans and DNNs on *VPT-basic* by measuring the correlations between their error patterns with Cohen's $\kappa$ (Geirhos et al., 2020b). Model $\kappa$ scores were mostly correlated with accuracy on *VPT-basic* after linear probes and fine-tuning (Fig. 9). However, while nearly all DNNs were highly correlated with human error patterns after fine-tuning, the correlation between $\kappa$ scores and task accuracy disappeared (Fig. 9B, purple dots).

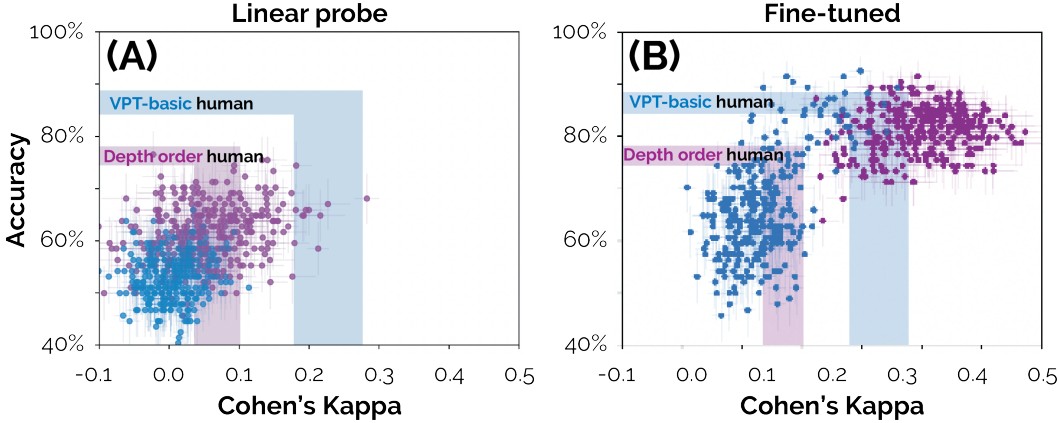

Figure 9: **Error pattern correlations (Cohen's κ) between humans and DNNs on *VPT-basic* and *Depth order*.**

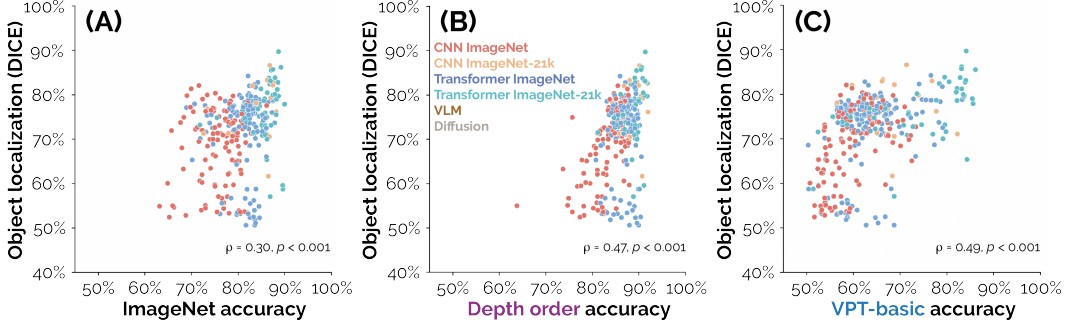

Figure 10: **DNNs fine-tuned on VPT-basic learn to solve the task by overfitting to the positions of the target objects. (A, B, C)** There are significant correlations between DNN accuracy on ImageNet, fine-tuned accuracy on Depth/VPT-basic tasks, and the extent to which their attribution maps localize the target objects in VPT. Given the dramatic failure of DNNs to solve *VPT-Strategy*, this finding implies that DNNs rely on a and brittle strategy of memorizing object features like size and position to solve VPT.

| Architecture | Model | Versions |
|---|---|---|
| CNN | AlexNet (Krizhevsky et al., 2012b) | 1 |
| | ConvMixer (Trockman & Kolter, 2022) | 3 |
| | ConvNeXT (Liu et al., 2022) | 10 |
| | DenseNet (Huang et al., 2018) | 4 |
| | DLA (Yu et al., 2019) | 5 |
| | DPN (Chen et al., 2017) | 6 |
| | EfficientNet (Tan & Le, 2020) | 4 |
| | GhostNet (Han et al., 2020b) | 1 |
| | HRNet (Sun et al., 2019) | 8 |
| | LCNet (Cui et al., 2021) | 3 |
| | MixNet (Tan & Le, 2019) | 4 |
| | MnasNet (Tan et al., 2019) | 3 |
| | MobileNet (Howard et al., 2019) | 14 |
| | RegNet (Radosavovic et al., 2020) | 6 |
| | Res2Net (Gao et al., 2021) | 5 |
| | ResNet (He et al., 2015) | 26 |
| | ResNeSt (Zhang et al., 2020) | 3 |
| | RexNet (Han et al., 2020a) | 5 |
| | ResNext (Xie et al., 2016) | 2 |
| | SPNASNet (Stamoulis et al., 2019) | 1 |
| | TinyNet (Han et al., 2020c) | 2 |
| | VGG (Simonyan & Zisserman, 2014) | 14 |
| Transformer | BEiT (Bao et al., 2021b) | 9 |
| | CAFormer (Yu et al., 2022b) | 6 |
| | CaiT (Touvron et al., 2021b) | 3 |
| | ConViT (d'Ascoli et al., 2021) | 3 |
| | CrossViT (Chen et al., 2021a) | 2 |
| | DaViT (Ding et al., 2022) | 3 |
| | DeiT (Touvron et al., 2021a) | 12 |
| | EfficientFormer (Li et al., 2022b) | 7 |
| | EVA (Fang et al., 2023) | 9 |
| | FocalNet (Yang et al., 2022) | 6 |
| | LeViT (Graham et al., 2021) | 5 |
| | MaxViT (Tu et al., 2022) | 6 |
| | MobileViT (Mehta & Rastegari, 2022) | 3 |
| | MViT (Li et al., 2022a) | 3 |
| | PiT (Heo et al., 2021) | 8 |
| | PVT (Wang et al., 2022) | 7 |
| | Swin (Liu et al., 2021) | 16 |
| | Twins-SVT (Chu et al., 2021) | 5 |
| | ViT (Dosovitskiy et al., 2021b) | 36 |
| | Volo (Yuan et al., 2022) | 7 |
| | XCiT (El-Nouby et al., 2021) | 6 |
| | PoolFormer (Yu et al., 2022a) | 8 |
| Hybrid | CoaT (Xu et al., 2021) | 7 |
| | CoAtNet (Dai et al., 2021) | 8 |
| | EdgeNeXt (Maaz et al., 2022) | 1 |
| | Visformer (Chen et al., 2021b) | 2 |
| Foundation | Depth Anything (Yang et al., 2024) | 1 |
| | DINOv2 (Oquab et al., 2023) | 1 |
| | iBoT (Zhou et al., 2021) | 1 |
| | MAE (He et al., 2022) | 1 |
| | MiDas (Ranftl et al., 2022) | 1 |
| | SAM (Kirillov et al., 2023) | 1 |
| VLM | ChatGPT4 (Achiam et al., 2023) | 1 |
| | Gemini (Team et al., 2023) | 1 |
| | Claude 3 (Anthropic, 2024) | 1 |
| Diffusion | Stable Diffusion 2.0 (Rombach et al., 2021) | 1 |

Table 1: **The 327 DNN models used in our study.**

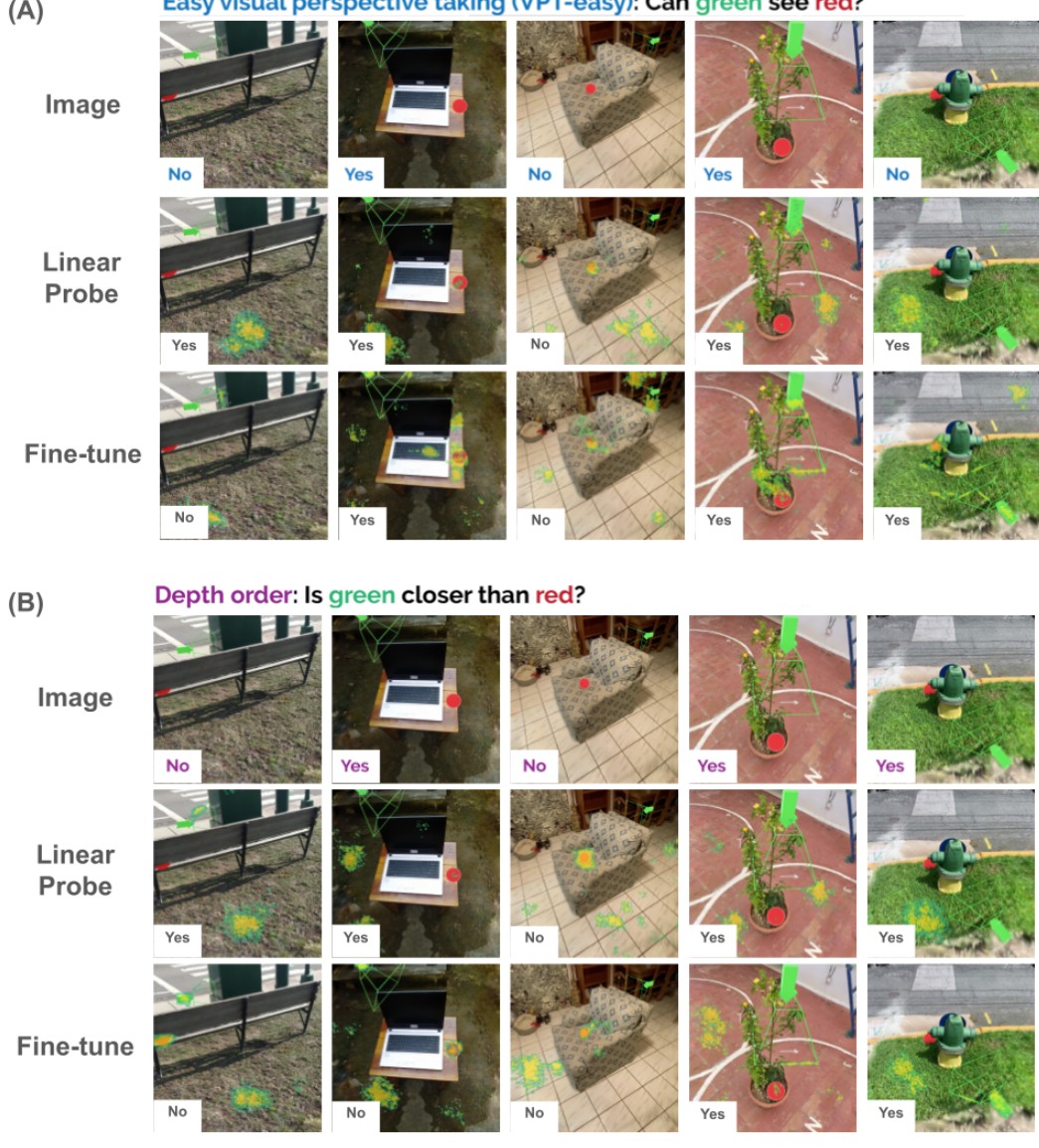

Figure 11: **Examples of decision attribution maps from a `ViT large`. (A)** Attribution maps from linear probed and fine-tuned DNNs on VPT-basic. **(B)** Attribution maps from linear probed and fine-tuned DNNs on depth order.

