# OpenReview forum: "The 3D-PC: a benchmark for visual perspective taking in humans and machines"
_ICLR.cc/2025/Conference — ICLR 2025 Poster_

### Official Review · Reviewer_gKTT · 2024-11-03

**Soundness:** 3
**Presentation:** 3
**Contribution:** 3
**Rating:** 6
**Confidence:** 3

**Summary:**

This paper introduces the 3D-PC benchmark, a new evaluation framework for visual perspective-taking (VPT) in humans and DNNs, focusing on spatial understanding and social interactions. The benchmark consists of three tasks: Object Depth Ordering, Basic VPT, and VPT-Perturb, where the latter is designed to prevent DNNs from using shortcut strategies. Through evaluation of 33 human participants and over 300 DNNs, the results show that while DNNs achieve human-comparable performance in depth ordering, they show clear limitations in VPT tasks, particularly VPT-Perturb. While fine-tuning helps DNNs match human performance on Basic VPT, they perform at chance level on VPT-Perturb, indicating a gap in spatial reasoning generalization. The findings demonstrate limitations in DNNs trained on static image datasets for 3D reasoning, suggesting current architectures lack the visual strategies needed for human-like perspective-taking.

**Strengths:**

The paper has good insight about cognitive understanding of 3D perception, exploring the gap between human and DNN capabilities in perspective-taking. By demonstrating that DNNs trained on static images struggle with tasks requiring true 3D reasoning (notably in VPT-basic and VPT-perturb as shown in Figures 4 and 6), the authors highlight a significant limitation of current model approaches. The paper also nicely suggests incorporating cognitive science insights as a potential path forward.

**Weaknesses:**

1. While the paper presents interesting tasks for studying 3D understanding, it could benefit from broader connections to the current 3D computer vision landscape.

2. Adding some discussion about how their findings relate to work in depth estimation and 3D segmentation would help put their cognitive insights into a wider context.

3. Figure 6 provides an overview of DNN performance but deeper exploration about why models fail on VPT tasks would be better. Including detailed failure cases or decision attribution maps could shed light on where models diverge from human performance, particularly in VPT-perturb where models rely on “shortcut” strategies.

**Questions:**

See weakness for more information about potential questions and improvements.

---

> ### Author Response · Authors · 2024-11-23
> **Response**
>
> **While the paper presents interesting tasks for studying 3D understanding, it could benefit from broader connections to the current 3D computer vision landscape… Adding some discussion about how their findings relate to work in depth estimation and 3D segmentation would help put their cognitive insights into a wider context**
> We appreciate this feedback. Our original manuscript focused more on the growing body of work describing how 3D spatial property perception capabilities have begun to emerge in image analysis models. But we agree that we could expand the context to 3D computer vision more broadly. We have expanded our related work to incorporate this context.
>
> **Figure 6 provides an overview of DNN performance but deeper exploration about why models fail on VPT tasks would be better. Including detailed failure cases or decision attribution maps could shed light on where models diverge from human performance, particularly in VPT-perturb where models rely on “shortcut” strategies.**
> We included comparisons of DNN decisions and error patterns (using Cohen’s Kappa) in section 9 of the appendix (also see Fig. 8). These experiments showed strong linear relationships for both depth order and VPT-basic between DNN accuracy and human alignment on these tasks. To better understand DNN failures on VPT-strategy, we compared DNN decision attribution maps to the ground truth location of objects in each scene (Fig. 9). DNNs do not learn to estimate the line-of-sight of target objects to solve VPT; instead, as this experiment suggests, they become progressively more sensitive to object features and their spatial relationships as they improve on VPT-basic (Fig. 9C). To expand on these analyses, we have added example DNN attribution maps for depth order, VPT-basic, and VPT-strategy to the Appendix of our updated submission. Also see our **Response to all reviewers** for additional thoughts.

---

> > ### Comment · Area_Chair_CW8g · 2024-11-25
> >
> > Dear reviewer,
> >
> > The authors have provided another round of responses. Could you kindly review them and provide your feedback?
> > Thanks

---

> ### Comment · Reviewer_gKTT · 2024-11-27
> **Response**
>
> I appreciate the authors' responses to my questions and their added details to the paper. My positive assessment remains unchanged.

---

### Official Review · Reviewer_aZ4R · 2024-11-04

**Soundness:** 1
**Presentation:** 3
**Contribution:** 2
**Rating:** 6
**Confidence:** 4

**Summary:**

This paper proposes 3D-PC, a new benchmark for visual perspective taking (VPT), an ability to predict what other perspectives see. The benchmark consists of three tasks: 1) telling which object is closer (depth order), 2) predicting if a camera can see a ball (VPT-basic), 3) another version of 2) where the occlusion is flipped. The authors then tested 33 human participants and over 300 DNNs on the benchmark, showing that despite a super-human performance on depth order, DNNs are near chance on VPT tasks.

**Strengths:**

- The proposed benchmark is clearly motivated and well situated in the literature.
- The methodology is clearly explained with nice figures.
- Extensive experiments are conducted to show how the challenge can boost the 3D perception research of deep learning.

**Weaknesses:**

- I think for a benchmark paper, one of the most important things is the correctness of the ”ground truth labels”. Although the method proposed may sound reasonable, it is hard to tell how unlikely the “GTs” are inaccurate. After all, the adopted 3D Gaussian Splatting method is not perfect. Therefore, I expect some experiments or at least some discussions on the correctness of GTs.
- I think the descriptions for VPT-Strategy are insufficient. I could only understand what this task was about when I read Figure 6.
- I don’t think we can say that DNNs learn depth from the depth order experiments. It seems to be very likely that the linear-probed DNNs only learned to solve the task based on the relative sizes of the red ball and green camera. Therefore, I don’t see the meaning of the depth order task.
- It seems that the role of the VPT-Strategy task is to test if the observer estimates the line-of-sight of the camera. However, I think it cannot take such a role. I guess if you finetune the DNNs on VPT-Strategy data, they will perform much better. If so, we can’t say they are estimating the line-of-sight because they are likely still using features to perform the task. Therefore, I don’t think the VPT-Strategy experiments are very meaningful.

Minor
- It is claimed that AlexNet is included in the model zoo (L250), but I didn’t find it in Appendix Table 1?
- Figure 2: VPT-easy → VPT-basic.

**Questions:**

Please see the Weaknesses. Specifically, please answer:

- How do you evaluate the correctness of the ground truth labels generated by the proposed method?
- Can you do some experiments to show whether or not the DNNs learn to perform the depth order task based on relative sizes?
- How will the DNNs perform if they are finetuned on VPT-Strategy data? Do you think the results may change your perspective on the VPT-Strategy’s role?

---

> ### Author Response · Authors · 2024-11-23
> **Response**
>
> **I think for a benchmark paper, one of the most important things is the correctness of the ”ground truth labels”. Although the method proposed may sound reasonable, it is hard to tell how unlikely the “GTs” are inaccurate. After all, the adopted 3D Gaussian Splatting method is not perfect. Therefore, I expect some experiments or at least some discussions on the correctness of GTs.**
>
> Our goal was to systematically evaluate the 3D perception capabilities of DNNs and humans. We were particularly interested in understanding if DNNs could do VPT, since this is an ability that emerges over the first decade of life in humans. We generated images and derived ground truth labels using Gaussian Splatting in 3D space to ensure correctness. Ultimately, the true test for us was how well human participants could solve depth order and VPT in those images. In each case, humans were significantly above chance (Fig. 2, 75% accurate on depth order, 87% on VPT-basic, and 87% on VPT-strategy). Also note that DNNs *outperformed* humans on depth order after fine-tuning (Fig. 5A and 5C). In summary, humans (and models) were able to solve multiple 3D tasks posed on these images, indicating that our Gaussian Splatting image generation approach was sufficient for generating accurate GT labels. The failure of DNNs to solve VPT is not a byproduct of Gaussian Splatting artifacts.
>
> **I think the descriptions for VPT-Strategy are insufficient. I could only understand what this task was about when I read Figure 6.**
> Thanks for this comment. We have revised our explanation in our updated manuscript.
>
> **I don’t think we can say that DNNs learn depth from the depth order experiments. It seems to be very likely that the linear-probed DNNs only learned to solve the task based on the relative sizes of the red ball and green camera. Therefore, I don’t see the meaning of the depth order task.**
> There are multiple cues that humans rely on for perceiving object and scene depth, including the relative sizes of objects. As pointed out by the reviewer, relative sizes of objects are also likely a strong cue for solving our depth order task. The fact that DNNs can utilize this (and other) cues after linear probing is evidence that they learn human-like strategies for perceiving 3D spatial properties. As an example of the importance of relative size for 3D vision in humans, consider the Ames illusion, in which a person appears far bigger than they actually are based on the viewer’s expectations about their size. See our **Response to all reviewers
> ** for more details about monocular depth cues, relative size as a candidate cue for solving our depth order task, and why comparing performance on VPT vs. depth order on the exact same images enables unique and significant insights into differential 3D perception capabilities of humans vs. DNNs.
>
> **It seems that the role of the VPT-Strategy task is to test if the observer estimates the line-of-sight of the camera. However, I think it cannot take such a role. I guess if you finetune the DNNs on VPT-Strategy data, they will perform much better. If so, we can’t say they are estimating the line-of-sight because they are likely still using features to perform the task. Therefore, I don’t think the VPT-Strategy experiments are very meaningful.**
>
> The purpose of VPT-strategy was to see if DNNs (and humans) could solve VPT as we linearly moved the objects through the environment with a fixed camera. This should be a trivial perturbation for an observer that estimates the line-of-sight of the green camera, and indeed it is for humans. However, DNNs do not use line-of-sight, and hence struggle to generalize on images with object perspectives that rarely or never appear in the training set. To summarize, line-of-sight is not a necessary strategy for learning to solve VPT-basic, but it is necessary to generalize from VPT-basic to VPT-strategy.
>
> **It is claimed that AlexNet is included in the model zoo (L250), but I didn’t find it in Appendix Table 1?**
> We apologize for the oversight. We have added AlexNet in Appendix Table 1.
>
> **Figure 2: VPT-easy → VPT-basic.**
> Thank you for pointing this out. We have fixed it in our revision.

---

> > ### Comment · Reviewer_aZ4R · 2024-11-24
> > **Thank you for your response**
> >
> > > In summary, humans (and models) were able to solve multiple 3D tasks posed on these images, indicating that our Gaussian Splatting image generation approach was sufficient for generating accurate GT labels. The failure of DNNs to solve VPT is not a byproduct of Gaussian Splatting artifacts.
> >
> > I cannot agree with the statements. The fact that the accuracies of humans and models (after fine-tuning) are above chance is not sufficient to prove that GT labels are accurate because there could be a part of the inaccurate labels that make them answer wrongly (or mistakenly correctly).
> >
> > > We have revised our explanation in our updated manuscript.
> >
> > I'm sorry but I don't see the updated explanations. Can you please point it out for me?
> >
> > > ...relative sizes of objects are also likely a strong cue for solving our depth order task. The fact that DNNs can utilize this (and other) cues after linear probing is evidence that they learn human-like strategies for perceiving 3D spatial properties.
> >
> > I think the key is there is no evidence that DNNs are using *other* cues than relative sizes to perform depth order task. Is there any? Also, I don't think relative sizes should be called "cues" in this case, or at least shouldn't be understood as it is in the human case. For example, humans know a dog is usually smaller than an adult, so when the dog is visually much larger than the adult, humans tend to believe that the dog is much closer to the camera/eyes. But in the case of depth order, there is no such prior knowledge involved.
> >
> > > As an example of the importance of relative size for 3D vision in humans, consider the Ames illusion, in which a person appears far bigger than they actually are based on the viewer’s expectations about their size.
> >
> > I think the example of the Ames illusion demonstrates exactly the opposite of what you emphasised, i.e., it shows how relative size is vulnerable for humans to use as a cue for estimating depth. It is because other cues are so powerful over relative size that humans tend to believe a person's real size gets bigger as walking.
> >
> > > To summarize, line-of-sight is not a necessary strategy for learning to solve VPT-basic, but it is necessary to generalize from VPT-basic to VPT-strategy.
> >
> > I don't see why line-of-sight is necessary to perform VPT-strategy. As I said, I believe if you finetune the DNNs on VPT-Strategy data, they will perform much better, but they are likely still using features to perform the task.

---

> > > ### Author Response · Authors · 2024-11-24
> > > **Appreciate the feedback. Response (1/3)**
> > >
> > > We really appreciate the reviewer’s feedback and the chance to continue to improve our work.
> > >
> > > **I cannot agree with the statements. The fact that the accuracies of humans and models (after fine-tuning) are above chance is not sufficient to prove that GT labels are accurate because there could be a part of the inaccurate labels that make them answer wrongly (or mistakenly correctly).**
> > >
> > > This is a critical point, so we appreciate the opportunity to clarify and better understand the reviewer’s concern. We designed our stimuli in 3D with gaussian splatting-generated scenes and Unity, then generated 2D views that we tested humans and DNNs on. Objects were staggered in 3D w.r.t. the viewing camera's position, and we adjusted the rotation of the green camera so that it was/wasn’t pointing at the red object and the line-of-sight was/wasn’t occluded by another object in the scene (see **Methods**, the 3D-PC). We can expand our description of our stimulus design pipeline if the reviewer thinks it would help clarify these issues.
> > >
> > > Our stimuli had correct 3D information in the stimuli by design (3D Gaussian Splatting models and all images can be found at https://huggingface.co/datasets/3D-PC/3D-PC). As mentioned, the important validation for us was to see if humans could perceive the depth order and VPT of the objects in these scenes. The fact that human participants could solve these tasks on these stimuli, and that their responses and errors were highly correlated (see Fig. 1 for accuracies Fig. A.9 for error pattern correlations) meant that the visual cues and routines that underlie human 3D vision were effective for solving depth order and VPT tasks on our stimuli.
> > >
> > > **I think the key is there is no evidence that DNNs are using other cues than relative sizes to perform depth order task...**
> > > As discussed in our response above, we designed the stimuli by manipulating the positions and poses of the target objects in a 3D scene. Since humans were able to determine the depth order of objects in these images, it meant that they leveraged monocular depth cues. Monocular depth cues for humans in static images are the relative size of objects, their lighting and shadows, their relative heights, linear perspective, occlusion/interposition, and texture gradients. We believe (as the reviewer originally pointed out) that relative size is an especially strong cue for determining depth order in these images. Relative size could be learned by humans over the training period that familiarized them with the task (see **Psychophysics experiment**) and by DNNs through linear probes or fine-tuning. Another cue that could be used by both is object occlusion/interposition. See Fig. 2 for examples (such as the chair in the bottom right) of occlusions in the dataset and how that can help solve depth order.
> > >
> > > Our goal in this study wasn’t to drill down on any specific monocular depth cue, although that is a future line of work we are very interested in. Instead, our goal was to understand if humans/DNNs could achieve similar accuracy and decision patterns on this challenge. So we are agnostic to the precise depth cues or combinations of cues that humans/DNNs used. However, we take the reviewer’s point to be a fair one, and we have added the following limitations section to our manuscript (that we have also just uploaded):
> > >
> > > >One key limitation of our study is that we were not able to isolate the precise cues that humans versus DNNs use to solve \textit{depth order}. While both were very capable of solving the task, it's very possible that both relied on different monocular depth cues. For example, the relative size of objects is a very salient cue for solving the task. However, object occlusion and interposition is another strong cue in this dataset, and it's unclear if DNNs and humans rely on one or the other, or both, or other cues like the lighting and shadows of objects. Future work that can drill down on the strategies of humans and DNNs for perceiving 3D spatial properties will make significant contributions to understanding the alignment of biological and machine vision.
> > >
> > > > Another key limitation of our study is that our version of VPT represents the most basic form studied in the developmental psychology literature. While solving this task is evidently an extraordinary challenge for DNNs, it is only one small step towards human-level capabilities for reasoning about 3D worlds in general. Far more research is needed to identify additional challenges, architectures, and training routines that can help DNNs perceive and reason about the world like humans do. We release our \texttt{3D-PC} data, 3D Gaussian Splatting models, and code anonymously at \url{https://huggingface.co/datasets/3D-PC/3D-PC} to support this goal.

---

> > > > ### Comment · Reviewer_aZ4R · 2024-11-25
> > > > **My main concerns remain but I am raising my score**
> > > >
> > > > Thank you very much for your detailed and in-depth response. I still have two main concerns:
> > > >
> > > > 1) You describe again the data generation process and then claim "Our stimuli had correct 3D information in the stimuli by design". But I don't see how the "GT" is correct by design since the adopted 3D Gaussian Splatting method is imperfect. It also contradicts your earlier response ("Ultimately, the true test for us was how well human participants could solve depth order and VPT in those images"). I have stated the reason why I don't think it's the "ultimate true test".
> > > >
> > > > 2) I still believe the the 3D structure of the images is irrelevant in the depth order task which is only about the relative sizes of the green camera and the red ball. I came up with a simple experiment design that can verify this assumption: ablate the images, place the camera and ball in a vacuum while keeping their relative position the same, and see if human and fine-tuned models perform much worse. If not, it suggests that depth order can be performed only with relative sizes, and in this case, it shouldn't be seen as monocular depth cues anymore because, for a binary classification model, the question now becomes "Which object appears larger?".
> > > >
> > > > Despite the concerns above, I think my other questions are well addressed, and I think the VPT tasks are interesting and would contribute positively to the 3D perception community. Therefore, I would like to raise my score to 6.

---

> > > > > ### Author Response · Authors · 2024-11-26
> > > > > **Response**
> > > > >
> > > > > Thanks again for this discussion. We believe the reviewer's comments have really helped us improve our manuscript.
> > > > >
> > > > > **I still believe the the 3D structure of the images is irrelevant in the depth order task which is only about the relative sizes of the green camera and the red ball. I came up with a simple experiment design that can verify this assumption: ablate the images, place the camera and ball in a vacuum while keeping their relative position the same, and see if human and fine-tuned models perform much worse. If not, it suggests that depth order can be performed only with relative sizes, and in this case, it shouldn't be seen as monocular depth cues anymore because, for a binary classification model, the question now becomes "Which object appears larger?".**
> > > > >
> > > > > We agree this is an interesting idea. To test it, we ran another experiment on the ViT Large model. We compared the model's performance on the standard depth order task with a version where the background was masked out. We compared performance on a subset of images with ground truth segmentation masks and found the following:
> > > > >
> > > > > | Condition | Image | Background Mask | Performance Difference |
> > > > > |-----------|-------|-----------------|------------------------|
> > > > > | Linear Probe | 69.23% | 57.33% | -11.9% |
> > > > > | Fine-tune | 97.44% | 85.33% | -12.11% |
> > > > >
> > > > > Our results reveal two key observations:
> > > > >
> > > > > 1. After linear probing the model for depth order, its performance dropped from 69.23% to 57.33% when the background was masked (-11.9 percentage points).
> > > > > 2. After fine-tuning the model for depth order, its performance decreased from 97.44% to 85.33% when the background was masked (-12.11 percentage points).
> > > > >
> > > > > These findings suggest that while relative size is a significant cue for depth order perception, it is not the only cue that the model relies on for depth order. We will include this result in the Appendix, as we agree with the reviewer that it is helpful for understanding our depth order results.

---

> > > > > > ### Comment · Reviewer_aZ4R · 2024-11-26
> > > > > >
> > > > > > Thank you for your response. I think the result suggests that although relative size is significant for depth order, other cues matter a lot as well, indicating 3D structure is relevant in the depth order task.

---

> ### Author Response · Authors · 2024-11-24
> **Response (2/3)**
>
> **I think the example of the Ames illusion demonstrates exactly the opposite of what you emphasised, i.e., it shows how relative size is vulnerable for humans to use as a cue for estimating depth. It is because other cues are so powerful over relative size that humans tend to believe a person's real size gets bigger as walking.**
> We agree with the reviewer and appreciate the chance to discuss this point. We only brought up the Ames illusion as an intuitive example of how the relative sizes of objects affect human depth perception. We apologize if this instead added any confusion.
>
> **I don't see why line-of-sight is necessary to perform VPT-strategy. As I said, I believe if you finetune the DNNs on VPT-Strategy data, they will perform much better, but they are likely still using features to perform the task.** We agree with the reviewer that DNNs could likely learn to solve VPT-strategy with sufficient fine-tuning and data. However, we designed VPT-strategy to be a strong generalization condition for observers originally trained on VPT-basic: it consists of the same objects as in VPT-basic but with different viewpoints and object configurations. VPT-strategy is therefore a hypothesis-driven approach to identify observers (DNNs or humans) that **do not** use line-of-sight for VPT. DNNs failed to generalize to this condition while humans were successful. We have added another paragraph to the limitations section of our Discussion to address the reviewer’s question.
> >Finally, we validated developmental psychology work that humans rely on a line-of-sight strategy to solve VPT, and demonstrated in our \textit{VPT-Strategy} task that DNNs learn to use a different strategy after fine-tuning. We provided evidence through our attribution map experiment (Appendix A.9) that DNN strategies are highly focused on the target objects, which means they may overly rely on the positions and sizes of target objects to solve the task. However, we were not able to characterize exactly how DNNs use object features for VPT, which may be critical for developing the next-generation of vision models that can solve VPT like humans do.

---

> > ### Author Response · Authors · 2024-11-24
> > **Response (3/3)**
> >
> > **I'm sorry but I don't see the updated explanations. Can you please point it out for me?** Sorry for not being clearer in our response. To address the reviewer’s original critiques about our descriptions of VPT-strategy, we rewrote the **DNNs learn different strategies than humans to solve VPT** section. See below for old and new versions.
> >
> > **Old version:**
> > >VPT-Strategy has observers solve the VPT task on a series of images rendered from a fixed camera viewpoint as the green camera and red ball are moved incrementally from one side of the screen to the other, passing by an occluding object in the process. This means …
> >
> > **New version:**
> > > \paragraph{DNNs learn different strategies than humans to solve VPT.} Prior work has found that Deep Neural Networks (DNNs) can achieve human-level performance on visual tasks while using fundamentally different strategies than humans. To investigate whether this issue applies to VPT, we designed a new experiment which we call \textit{VPT-Strategy}.
> > > \textit{VPT-Strategy} was inspired by past work in developmental psychology, where it has been proposed that humans solve VPT by estimating line-of-sight because humans respond in predictable ways when objects in a scene are slightly repositioned. In \textit{VPT-Strategy}, observers solve the VPT task on a series of images rendered from a fixed camera viewpoint which show the green camera and red ball incrementally repositioned from one side of the screen to the other, passing by an occluding object in the process. This makes it possible to precisely map out the moments at which the green camera's view of the red ball is unoccluded, then occluded, then unoccluded once more. DNNs that have been fine-tuned on \textit{VPT-basic} behaved differently than humans on this task: humans were 87\% accurate, but the highest performing DNN, the \texttt{Swin Transformer} trained on ImageNet-21k, was only 66\% accurate. Since DNNs cannot generalize to \textit{VPT-Strategy}, it means that they do not learn to estimate line-of-sight to solve VPT.
> > > To better understand the source of DNN failures, we derived ``smooth gradients'' decision attribution maps of every DNN that we tested on \textit{VPT-Strategy}, and then measured the overlap of these maps with the green camera and red ball objects using Dice similarity coefficient. These attribution maps revealed that the DNNs we evaluated did indeed learn to attend to the locations of the green camera and red ball objects when trying to solve VPT, and the precision of their localization increased as a function of accuracy on \textit{VPT-basic}. When considered alongside the poor DNN performance on \textit{VPT-strategy}, these results imply that DNNs learn a brittle feature-based strategy that is dependent on object properties like size and location instead of estimating the line-of-sight of the green camera like humans do.

---

> > > ### Comment · Area_Chair_CW8g · 2024-11-25
> > >
> > > Dear reviewer,
> > > The authors have provided another round of responses. Could you kindly review them and provide your feedback?

---

### Official Review · Reviewer_NwpN · 2024-11-06

**Soundness:** 3
**Presentation:** 3
**Contribution:** 4
**Rating:** 8
**Confidence:** 3

**Summary:**

Visual Perspective Taking (VPT) refers to our ability to understand how others see the world from their viewpoint. This paper has developed a new benchmark to test whether DNNs can perform this skill. Their test includes three components: determining the order of objects by depth, VPT-basic, and VPT-strategy that eliminates potential shortcuts solutions.

While DNNs matched or even exceeded human ability in determining object depth order, they struggled significantly with VPT-basic, performing at nearly random levels. In contrast, humans excelled at these same tasks with almost perfect accuracy. Even when DNNs are fine-tuned for VPT-basic, they still failed at VPT-Strategy.

These findings suggest that DNNs don't naturally develop the ability to take others' perspectives, even after extensive training on static images or fine-tuning on VPT-basic tasks.

**Strengths:**

- The paper used 3D Gaussian Splatting to create large sets of realistic images.
- This method gave them precise control over the test conditions, helping them mitigate confoudning factors.
- Their analysis revealed interesting insights: while DNNs can learn to understand depth from static images, they don't develop the ability to take others' perspectives. They also found that a network's accuracy on ImageNet correlates with the 3D capabilities of DNNs they tested.

**Weaknesses:**

Using only green cameras and red target points in their test images raises a few questions: Would the main results still hold if they used other colors or camera shapes? Is the paper assuming the DNNs can easily spot the camera and determine its viewing angle in the images? If so, how much do the camera's physical properties (like its size, line thickness, or color) affect the results?

**Questions:**

The paper mentions in its appendix that when testing Visual Language Models (VLMs), they provided in-context examples to help guide the models. However, it's unclear whether they also included step-by-step reasoning examples like chain-of-thought prompting, which can probably help the models better understand how to approach these perspective-taking tasks.

---

> ### Author Response · Authors · 2024-11-23
> **Response**
>
> **Using only green cameras and red target points in their test images raises a few questions: Would the main results still hold if they used other colors or camera shapes? Is the paper assuming the DNNs can easily spot the camera and determine its viewing angle in the images? If so, how much do the camera's physical properties (like its size, line thickness, or color) affect the results?**
> We decided to use a red ball and green camera as our target objects because they are visually salient and easy to see. Indeed, humans were significantly above chance accuracy on depth order and both VPT tasks (Figs 3, 4, and 6). We also found no evidence that DNNs struggled to see the objects: all DNNs linearly probed on the depth order task performed significantly above chance (all were at least *p < 0.05*), and the best models were even more accurate than humans (Fig. 4C). In other words, we found no evidence that the colors or shapes of the target objects were problematic.
>
> **The paper mentions in its appendix that when testing Visual Language Models (VLMs), they provided in-context examples to help guide the models. However, it's unclear whether they also included step-by-step reasoning examples like chain-of-thought prompting, which can probably help the models better understand how to approach these perspective-taking tasks.**
> Our goal in our original submission was to compare humans and VLMs on depth/VPT as fairly as possible. As discussed in A.6, we gave VLMs the exact same training images and instructions as human participants. Despite this, they struggled immensely on VPT even after trying multiple response temperatures and taking the best possible accuracy.
>
> However, the reviewer raises a great point: if VLMs could achieve good accuracy on VPT after chain-of-thought prompting, it would imply that the challenge is more of a reasoning one than a perceptual one. To test this, we ran an additional experiment with GPT-4-Turbo, in which we cued it for chain-of-thought reasoning (see below for example depth and VPT instructions). However, chain-of-thought did not help the model’s performance (it was near chance accuracy; see Appendix A.7 for more details).
>
> Depth:
> Here is a sample training image, from the perspective shown, is the red ball closer to the observer or is the green arrow closer to the observer? Answer only 'BALL' if the red ball is closer, or 'ARROW' if the green arrow is closer, nothing else. To answer this, let's think step by step: The red ball is above the suitcase, which is a small distance away from the observer, approximately 3 feet. The green arrow is in front of the suitcase, which is closer to the observer, approximately 1 foot away. Thus the answer is: ARROW
>
> VPT:
> Here is a sample training image, if viewed from the perspective of the green 3D arrow, in the direction the arrow is pointing, can a human see the red ball? Answer only 'YES' or 'NO', nothing else. Let's think step by step: If a human were looking in the direction the arrow was pointing, they would face a few possible occlusions, such as the suitcase. The red ball is above the suitcase, so the suitcase wouldn't block the view of the red ball. Thus, the answer is: YES

---

> > ### Comment · Area_Chair_CW8g · 2024-11-25
> >
> > Dear reviewer,
> > The authors have provided their responses. Could you kindly review them and provide your feedback?

---

> > ### Comment · Reviewer_NwpN · 2024-11-26
> >
> > Thank you for your responses, it clarifies my questions. Also it's nice to see the additional experiment using chain-of-thought. I'm increasing my score.

---

### Author Response · Authors · 2024-11-23
**Response to all reviewers**

We thank the reviewers for their extensive feedback. We are confident that we have addressed the main critiques, which we summarize below along with our responses:

**Questions about the depth-order task (zZ4R, gKTT):** Humans rely on a number of monocular depth cues, such as linear perspective, texture gradients, occlusions, and relative size. As pointed out by the reviewers, the relative size of the green/red target objects is a strong cue for our depth-order task. DNN attribution maps indicate that they focus on both objects when solving the task, and that this focus increases as a function of performance on the depth-order task (Appendix Fig. 9b). Moreover, in our original submission we reported that models which were more accurate on the depth-order task were also more aligned with humans in their error patterns on the task (Appendix Fig. 8, purple dots, linear probe: $\rho=0.23, p < 0.001$, finetune: $\rho=0.24, p < 0.001$). In summary, the models in our zoo focused on object features — and potentially their relative sizes — to solve the depth-order task. However, relative size is also a cue that humans rely on for depth perception, and models and humans are strongly correlated in their decision making on the task.

More to the point raised by some of the reviewers, the purpose of including a depth-order task in the 3D-PC was to enable a broad comparison of the 3D perception capabilities of models and humans. We counterbalanced labels of depth-order and VPT-basic, and showed a strong dissociation between human 3D perception and 3D perceptual capabilities of nearly every kind of image-based model in use today — from ViTs to diffusion models and from CNNs to vision language models.

We believe that presenting the VPT results alone would be an incomplete snapshot of DNN 3D capabilities. Indeed, one of the most surprising results in our study is that DNNs, trained on static images, are growing progressively better at depth-order as they improve in object classification. In contrast, while there is a similar trend on VPT, the slope of this improvement is significantly smaller than on depth-order, which implies that a different modeling paradigm is needed for human-level VPT but not for other 3D vision tasks like depth-order. We discussed these findings and their connection to biological vision in our original submission (Discussion and Related work), and based on reviewer feedback, we have expanded our related work to connect to more current 3D computer vision work.

**Why do DNNs struggle at VPT (NwpN, zZ4R, gKTT)?** We demonstrated that DNNs combined with linear probes could not solve VPT-basic. Even after fine-tuning on VPT-basic, models could not generalize to new views where a human-like “line-of-sight” strategy would work (VPT-Strategy). In our original submission, we included an experiment to understand what strategy DNNs end up learning from fine-tuning (Appendix 9). Specifically, we computed attribution maps on every image for each model, and then looked at the overlap of these maps with the target objects (red ball/green camera). Plotting VPT-basic accuracy against object localization shows a significant correlation (Fig. 9c, $\rho=0.49, p < 0.001$): models learn to solve the task by becoming more exquisitely tuned to the object locations, but without learning to estimate line-of-sight between them; a strategy which we referred to as “feature-based” (Fig. 6A). Thus, our findings show that new architectures and/or data diets and/or training routines are needed for DNNs to learn a human-like strategy and achieve human-level performance on VPT.

---

### Meta-Review · Area_Chair_CW8g · 2024-12-12

**Metareview:**

This paper has developed a new benchmark to test whether DNNs can perform Visual Perspective Taking (VPT). Rigorous experiments reveal new insights that DNNs fail at VPT tasks.

The paper is well-written. The methodology for studying the problem and the benchmark are novel and interesting. The insights obtained from this study are useful for the future development of AI models in 3d depth perception.

Main concerns on depth order and the explanations why models struggle at VPT are addressed during the rebuttal.

All the reviewers reached an agreement to accept the paper after the rebuttal.

**Additional Comments On Reviewer Discussion:**

The authors have addressed the main concerns raised by the reviewers regarding depth ordering and the challenges models face in Visual Perceptual Tasks (VPT). Their explanations provided during the rebuttal effectively clarified these issues.

Additionally, specific questions raised by the reviewers—such as the need for a more detailed description of VPT, the verification of ground truth labels, and a discussion on the broader impacts of the work on 3D computer vision—were adequately addressed. The authors also expanded on related works in 3D segmentation, demonstrating a thorough understanding of the context of their work. These responses were satisfactory and contributed to resolving the reviewers' initial concerns.

---

### Decision · Program_Chairs · 2025-01-22

Accept (Poster)